# Phenotypic and Transcriptomics Analyses Reveal Underlying Mechanisms in a Mouse Model of Corneal Bee Sting

**DOI:** 10.3390/toxins14070468

**Published:** 2022-07-08

**Authors:** Yanzi Wang, Honghua Kang, Mengyi Jin, Guoliang Wang, Weifang Ma, Zhen Liu, Yuhua Xue, Cheng Li

**Affiliations:** 1Eye Institute & Affiliated Xiamen Eye Center, School of Medicine, Xiamen University, Xiamen 361102, China; 24520190154797@stu.xmu.edu.cn (Y.W.); 24520191153674@stu.xmu.edu.cn (H.K.); 24520181154690@stu.xmu.edu.cn (M.J.); liuzhen3394@xmu.edu.cn (Z.L.); 2Fujian Provincial Key Laboratory of Ophthalmology and Visual Science, School of Medicine, Xiamen University, Xiamen 361102, China; 3School of Pharmaceutical Sciences, Xiamen University, Xiamen 361102, China; 32320190154757@stu.xmu.edu.cn; 4Department of Ophthalmology, No.4 West China Teaching Hospital, Sichuan University, Chengdu 610041, China; 24520161153726@stu.xmu.edu.cn

**Keywords:** bee venom, corneal bee sting, hub gene, ocular surface damage, RNA-seq, transcriptome

## Abstract

Corneal bee sting (CBS) is one of the most common ocular traumas and can lead to blindness. The ophthalmic manifestations are caused by direct mechanical effects of bee stings, toxic effects, and host immune responses to bee venom (BV); however, the underlying pathogenesis remains unclear. Clinically, topical steroids and antibiotics are routinely used to treat CBS patients but the specific drug targets are unknown; therefore, it is imperative to study the pathological characteristics, injury mechanisms, and therapeutic targets involved in CBS. In the present study, a CBS injury model was successfully established by injecting BV into the corneal stroma of healthy C57BL/6 mice. F-actin staining revealed corneal endothelial cell damage, decreased density, skeletal disorder, and thickened corneal stromal. The terminal-deoxynucleotidyl transferase mediated nick end labeling (TUNEL) assay showed apoptosis of both epithelial and endothelial cells. Gene Ontology (GO) and Kyoto Encyclopedia of Genes and Genomes (KEGG) enrichment analysis showed that cytokine–cytokine interactions were the most relevant pathway for pathogenesis. Protein–protein interaction (PPI) network analysis showed that IL-1, TNF, and IL-6 were the most relevant nodes. RNA-seq after the application of Tobradex^®^ (0.3% tobramycin and 0.1% dexamethasone) eye ointment showed that Tobradex^®^ not only downregulated relevant inflammatory factors but also reduced corneal pain as well as promoted nerve regeneration by repairing axons. Here, a stable and reliable model of CBS injury was successfully established for the first time, and the pathogenesis of CBS and the therapeutic targets of Tobradex^®^ are discussed. These hub genes are expected to be biomarkers and therapeutic targets for the diagnosis and treatment of CBS.

## 1. Introduction

According to multiple surveys, hymenopterous insects, including ants, bees, and wasps, cause over 100 million injuries annually worldwide, among which bee stings account for almost 50% [1,2,3]. The majority of bee sting sites are focused on exposed skin of the head, face, and limbs. In mild cases, local redness, swelling, and pain may occur, while severe cases can present with severe allergic reactions, multiple organ dysfunction syndrome, and even death [4,5,6,7,8,9].

Previous studies often discussed bees (*Hymenoptera, Apoidea, Anthophila*) with two other insects, wasps and hornets, belonging to the same suborder Apocrita. They all have the character “Feng” in their Chinese names, which might be one of the reasons why they are often discussed together in China. Bees are so closely related to human life that the study of bees is extensive. However, researchers in different geographical locations have not reached a consensus because more than 16,000 species exist worldwide [10]. Therefore, we narrowed our focus to common bees. The most common bees in China are the Western honeybee (*Apis mellifera*) and the Chinese honeybee (*A. cerana*). Bee research has mostly focused on the Western honeybee, although the Western honeybee has about 33 subspecies [11]. Whether the components of bee venom (BV) secreted by these subspecies are identical has not been explored. Besides, the composition of bee venom also varies depending on bee age, geographical localization, seasonal changes, and social conditions [12]. This may be the reason why some differences exist in the results of the studies on bee venom components [12,13,14]. Therefore, the existence and proportion of each component are still doubtful.

However, the existence and composition of bioactive components in the BV are roughly the same. BV is a complex mixture of various active components, such as peptides (melittin, apamin, and mast cell degranulation (MCD) peptide), enzymes (phospholipase A2 (PLA2) and hyaluronidase), bioactive amines, and non-peptide components (histamine and catecholamines) [15]. Melittin is the most important compound present in BV and accounts for 50–60% of the contents; not only does it change the permeability of plasma membranes, causing cells to rupture, but it also affects the function of neurons, promoting the release of pain-inducing substances [16,17]. PLA2, which accounts for 10–12%, is a calcium-dependent enzyme with inflammatory effects that can cause hemolysis and disturb nociceptive-related cells [18]. Apamin, which accounts for 1–3% of BV, selectively inhibits Ca^2+^-dependent K^+^ channels (SK channels) in the central nervous system and activates inhibitory muscarinic M2 receptors in motor nerve endings [19,20]. Hyaluronidase can increase the diffusion rate of BV into tissues by disrupting them, leading to faster systemic changes [21,22]. MCD peptide, accounting for 1–2% of BV, induces histamine release from mast cells and plays an important role in inflammatory and allergic reactions [23].

As the initial barrier of the outermost layer of the eye, the cornea protects the intraocular tissue from external damage while transmitting light into the eye. It is divided into the epithelial layer, Bowman’s membrane, the stromal layer, Descemet’s membrane, and the endothelial layer [24]. Damage to any tissue layer in the cornea can lead to irreversible visual impairment. Although CBS is rare, it can cause a series of complications, including cicatricial conjunctivitis, corneal edema, corneal endothelial changes, iridocyclitis, anterior polar cataracts, and even chorioretinopathy. Tobradex^®^, a combination of steroids and antibiotics, is a widely used empirical therapy for eye infections. Studies have shown that it can significantly reduce inflammation of the eye [25]; however, due to the strong penetrating effect of BV and the complex ocular structure, it deteriorates even with a combination of topical glucocorticoid pulse therapy and anterior chamber irrigation [26,27,28,29].

Research into CBS has been underway since the end of the last century, but the pathological characteristics and pathogenesis have not been reported until recently. A variety of therapeutic strategies have been described for the treatment of CBS, although these are mainly in the form of individual case reports. Currently, apart from the establishment of pain models [30,31] and BV therapy [32,33,34,35,36], relevant animal studies are limited to skin surface irritation experiments. As the primary refractive medium of the eye, damage caused by blindness is self-evident. Consequently, CBS deserves attention in the form of positively exploring scientific and effective therapy methods. In the present study, we investigate the pathological characteristics and pathogenesis of CBS by establishing a stable and reliable animal model, which provides new therapeutic strategies for the clinical treatment of CBS and even skin bee stings.

## 2. Results

To verify the validity and stability of the corneal bee sting (CBS) model, we made a comparison with clinical cases. As shown in Figure 1 and Figure 2, significant ocular changes occurred in the BV group as compared with the Norm and VE groups. The ocular changes included corneal edema, pupil abnormalities, corneal epithelial defects, anterior chamber exudates, and cataract formation. Symptoms were persistent and resulted in corneal blindness and severe endophthalmitis, which are consistent with clinical cases and indicate successful construction of a stable and reliable CBS model.

Moreover, corneal edema and epithelial defects in the VT and BT groups were significantly improved as compared with the VE and BV groups, indicating successful establishment of a model of Tobradex^®^ treatment after a corneal bee sting.

### 2.1. Morphological Changes in the Cornea

The general morphological changes in whole cornea induced by CBS were evaluated. Corneal edema was severe in the BV group and mild in the VE, VT, and BV groups; however, the difference was not statistically significant (Figure 1A,C and Figure 2A,B). Corneal thickness increased significantly in the BV and BT groups; however, the corneas in the BT group were thicker than those in the BV group, which is an interesting phenomenon (Figure 2A,C and Figure 3C-Side view). Both corneal edema and thickness tended to increase and then decrease in the BV group, but only decreased over time in the BT group. These symptoms were not fully recovered at 72 h in the BV group but were absent at 72 h in the BT group. Moreover, observation of epithelial cell nuclei and cytoskeletal fluorescence staining demonstrated that corneal structure was significantly changed at 24 h after CBS. Furthermore, the typical corneal structure was no longer visible at 72 h after CBS in the BV group, with perfuse disordered nonspecific staining, while the corneal structure in the BT group was significantly changed but remained visible (Figure 3C,D view).

Next, changes in each layer of the cornea were evaluated. Corneal epithelium, the outermost layer of the cornea, can be directly observed and the changes quantitated by sodium fluorescein staining using slit-lamp microscopy. Epithelial defects only occurred in the BV and BT groups, and the area tended to decrease over time. These defects were not fully recovered at 72 h in the BV group but were absent at 72 h in the BT group (Figure 1B,D). Moreover, epithelial structure was significantly changed at 24 h after CBS in the BV group, and the typical epithelial structure was no longer visible at 72 h, with diffuse nonspecific staining. In the BT group, corneal epithelial cell structure was still visible in the areas with no defects (Figure 3A). Comparison with the VE group suggests that the corneal injury caused by CBS was due to both mechanical and toxic damage, which has been widely discussed in previous studies.

Corneal stroma, which occupies most of the thickness of the cornea, plays a major role in the thickening of corneal edema. Corneal speckle distribution and corneal cross-sectional area are often used to access the level and thickness of corneal edema. Corneal stroma had marked edema and was significantly thickened in the BV group, tending to increase and then decrease over time; however, these symptoms were not fully recovered at 72 h. Unexpected results were seen in the BT group, which are discussed later in combination with the whole-mount observations (Figure 2A–C). Observation of stromal nuclei and cytoskeletal fluorescence staining shows that typical stromal structure was no longer visible at 24 h in the BV group, and became more severe at 72 h, with perfuse disordered nonspecific staining. In the BT group, typical corneal stromal structure was still visible despite the altered stroma (Figure 3A), suggesting that corneal thickening and edema had different mechanisms in the BV and BT groups.

Corneal endothelium, the innermost layer of the cornea, has been widely studied because it plays a key role in corneal functional homeostasis. Corneal endothelium had disappeared completely at 24 h in the BV group, and had not recovered at 72 h. In the BT group, nuclear pyknosis was present at 24 h, and the cytoskeleton had disappeared; however, typical endothelial structure was observed at 72 h (Figure 3B). Moreover, the corneal endothelium is separated from the corneal stroma by a dense and impenetrable Descemet’s membrane, which is composed of collagen elastic fibers, indicating that BV can easily penetrate Descemet’s membrane and damage the corneal endothelium.

### 2.2. Morphological Changes in the Anterior Chamber

Aqueous humor in the anterior chamber plays a key role in the structural and functional stability of the anterior segment and can also reflect whether the function of the anterior segment is in the normal state. Anterior chamber exudate had accumulated by 24 h in the BV group, tending to increase and then decrease, and disappearing by 72 h after CBS, which indicates that CBS may induce acute iris-ciliary body dysfunction via the strong penetration ability of BV components. Although the accumulation of anterior chamber exudate also tended to increase and then decrease in the BT group, the degree of exudation was slightly weaker than that seen in the BV group at 6 h and 12 h; however, after 12 h, the degree of exudation was slightly stronger than that of the BV group and still existed at 72 h (Figure 2A,D). This suggests that the application of Tobradex^®^ slowed down the process of BV damage.

### 2.3. Apoptosis in the Cornea

In the BV group, signs of apoptosis were seen at 24 h. The structure of the three tissues had been destroyed by 72 h, indicating that the apoptotic process was complete. In the BT group, apoptosis in the epithelium and endothelium was seen at 24 h, suggesting that Tobradex^®^ slowed down the progression of damage elicited by BV (Figure 3A,B). Apoptosis in the VE and VT groups at 24 h may have been caused by mechanical damage, further indicating the double-injury factor of CBS. As a result of the data obtained from the BV group, statistical analysis of the number of apoptotic cells was not possible.

### 2.4. Transcriptomics Analysis in the BV Group

To ensure stability of the bee sting injury model, we performed correlation analysis between the VE and BV groups. The highest correlation coefficient was 0.990 within the sample groups and 0.723 between the groups, and the results (Figure 4A) indicate that our model was in a relatively stable state. To understand the intrinsic pathogenic mechanism of a bee sting, we next performed RNA-seq and analyzed the differentially expressed genes (DEGs) between the VE and BV groups. Principal component analysis (PCA) was performed to detect the degree of aggregation in the corneal samples. As shown in Figure 4B, 89.2% of the differences between the six corneal samples were from PC1; therefore, both the VE and BV samples were separated by PC1. According to the selection criteria of DEGs, the selected threshold was |log2(FC)| > 1 and *p*.adj < 0.05, and the number of individuals meeting this threshold was 3594, of which 1886 had upregulated expression (log2(FC) > 1) and 1708 had downregulated expression (log2(FC) < –1). DEGs are displayed as a volcano plot (Figure 4C). Subsequently, the hierarchical cluster analysis (HCA) of significantly up and downregulated genes was performed. A heatmap was plotted (Figure 4D) by https://www.bioinformatics.com.cn (accessed on 7 May 2022), a free online platform for data analysis and visualization showing the top 20 genes for each significant difference between the two groups.

To gain further insight into the differentially expressed genes, all 1886 upregulated DEGs were classified into GO functions: cellular components (CC) and molecular functions (MF). As shown in Figure 4E, BV injection triggered a series of biological processes such as leukocyte migration, T cell activation, ribosome biogenesis, and ribosomal RNA metabolic processes. The analysis of cellular components showed that the metabolic reactions occurred mainly in the membrane microdomain. Upregulated genes related to molecular function were mainly involved in integrin and DNA catalytic activity.

### 2.5. KEGG Pathway Enrichment Analysis in the BV Group

The KEGG metabolic pathways associated with environmental information processing and metabolism were significantly altered after CBS (Figure 5A). Further KEGG enrichment plots showed that the top 5 enriched pathways related to upregulated DEGs were cytokine–cytokine receptor interaction, viral protein–cytokine receptor interaction, osteoclast differentiation, small cell lung cancer, and DNA replication (Figure 5C). Interestingly, chemokine-related (CXCL3, CXCL5, CXCL11) and interleukin-related (IL-24, IL-1b, IL-12a) genes were consistent with the GO enrichment stress and chemokine responses (Figure 5B). Moreover, KEGG PATHVIEW showed that these chemokines were upregulated in the cytokine–cytokine receptor interaction pathway (Appendix A).

### 2.6. RNA-Seq Analysis following Tobradex^®^ Treatment of CBS

In clinical practice, the preferred treatment of bee stings is topical Tobradex^®^ application. To further study the underlying mechanism of action of Tobradex^®^, RNA-seq analysis of the BV and BT groups was performed. As shown in Figure 6A, 73.8% of the differences between the six corneal samples were from PC1; therefore, both the BV and BT samples were separated by PC1. A total of 2117 DEGs were identified by differential gene expression analysis, including 1639 upregulated genes and 478 downregulated genes (Figure 6A,B). Subsequently, the HCA of significantly up- and downregulated genes was performed. A heatmap was plotted (Figure 6C) by https://www.bioinformatics.com.cn (accessed on 7 May 2022), a free online platform for data analysis and visualization showing significant differences in gene expression patterns between the two groups.

GO enrichment analysis shows that upregulated DEGs were highly concentrated in visual perception, perception of light stimulation, regulation of transmembrane ion transport, organization of extracellular matrix, and extracellular structure. In terms of cell components, the extracellular matrix and collagen were enriched. Molecular functions mainly involved the activities of gated ion channels (Figure 6D,E). After Tobradex^®^ treatment of CBS, the main signaling pathways changed significantly, for example GABAergic synapses, morphine addiction, light signal transduction, nicotine addiction, and dopaminergic synapses (Figure 6F,G).

### 2.7. Genome-Wide Gene Set Enrichment Analysis (GSEA) and PPI Network

GO and KEGG analysis mainly investigated which pathways were enriched for DEGs and tend to miss some genes that are not significantly differentially expressed but are biologically important. Therefore, genome-wide GSEA enrichment analysis was performed to further investigate the functional alterations associated with BV. Figure 7A–C shows that the genome was most significantly enriched for reactome signaling by interleukins, reactome neutrophil degranulation, and reactome GPCR–ligand binding.

Hub genes are those that play a crucial role in biological processes and are influenced by the regulation of other genes in related pathways. These genes are often important targets in disease pathogenesis and treatment; therefore, the String database and Cystoscope plug-in Cyto-Hubba were used to construct protein–protein interaction network diagrams (PPI) composed of DEGs between the two groups. The degree of protein–protein interaction determines the size and color depth of each node. The top 30 and top 10 hub genes were screened with degree as 300 standards. As shown in Figure 7D–I, TNF, IL-6, and IL-1b were hub genes in the development of corneal injury caused by bee stings, and GFAP, MAG, and MBP played key roles in the mechanism of action of Tobradex^®^.

## 3. Discussion

Although bee stings are a common trauma seen in the emergency department, CBS is rare; thus, there are no in-depth studies or standardized therapy strategies for CBS to date. CBS leads to various ocular complications, such as corneal opacity, edema, corneal hyperalgesia, iris atrophy, secondary glaucoma, cataracts, endophthalmitis, and optic neuropathy, which can critically alter vision. Accordingly, early diagnosis and treatment of these complications are essential to preventing permanent blindness [37].

In the case of CBS, most patients present with corneal edema, corneal epithelial defects, photophobia, pain, and other inflammatory responses during the acute phase [38]. In the present study, 12 h after stromal injection of PBS, corneal edema subsided and the epithelial defects recovered rapidly; however, mice injected with BV still suffered to some extent from corneal epithelial defects and turbidity after 72 h. Moreover, even following treatment with Tobradex^®^, the cornea was thicker than that seen in the VE group, indicating incomplete healing [29]. Unexpectedly, residual corneal scarring and clouding of the anterior lens capsule is found in clinical cases several years after conventional treatment [39], and the density of endothelium in the affected eye is significantly reduced [29]. Therefore, it is imperative to investigate the pathogenesis of CBS with a view to exploring novel drug targets and biomarkers.

Corneal epithelium is the initial protective barrier of the eye against physical and chemical injury and pathogenic infection, thus maintaining transparency of the cornea and integrity of the visual system. Therefore, we hypothesized that even if the bee venom was extremely toxic, it could cause serious sustained damage to the cornea only after breaking through the epithelium and Bowman’s membrane. Then, the bee venom in the corneal stroma breaks through the Bowman’s membrane and enters the eye, causing damage to the whole anterior segment and even the whole eye. Therefore, we hypothesized that the penetration of bee venom through the Bowman’s membrane plays a key role in the process of CBS. Therefore, we used a needle to simulate the penetration of bee venom through the Bowman’s membrane and inject the bee venom into the corneal stroma to simulate CBS more realistically.

Corneal epithelial injury triggers cytokine-mediated interactions among epithelial, stromal, and immune cells. Damage to corneal epithelial cells leads to the release of a large number of cytokines (such as IL-1, IL-6, and TNF), which attract immune cells to migrate from limbal blood vessels to the corneal stroma. Destruction of Bowman’s membrane accelerates the diffusion of cytokines into the stroma and activates target cells. Mechanical damage to intraocular tissues caused by bee stings provokes a non-inflammatory cascade reaction, while melittin, PLA2, hyaluronidase, and other active substances in BV alter the permeability of the plasma membrane and accelerate the diffusion of BV, and histamine and MCD peptides trigger allergic reactions. The immune response in the cornea increases the release of chemokines, leading to the accumulation of inflammatory cells and ultimately resulting in corneal cell death.

CBS injuries are commonly associated with pain. The cornea is the most densely innervated tissue in the body and is supplied by the ciliary nerve from the ophthalmic branch of the trigeminal ganglion. Different types of sensory neurons are present in the cornea, which are functionally divided into those expressing multimodal nociceptive receptors, cold and heat receptors, and selective mechanonociceptive receptors [40,41]. Most of the sensory nerve fibers that innervate the cornea are of the multimodal type. Therefore, we hypothesized that the corneal pain caused by CBS might be attributed to the stimulation and destruction of corneal nerves by the bee venom.

Melittin activates transient receptor potential (TRP) channels through the PLA2 cascade, sensitizing primary pain receptors and causing pain [16]. According to GO and KEGG analysis of the differentially expressed and hub genes, we suggest that melittin and PLA2 work together on PLA2R1, thereby upregulating the expression of *PLA2G4C*, *NOS*, and *ALOX5AP* to generate metabolites such as NO and arachidonic acid. These metabolites, in turn, selectively activate the family of TRP channels (TRPV2, TRPV4, TRPM2, and TRPM6) through the phospholipase A2-lipoxygenase (PLA2-LOX) pathway, leading to Ca^2+^ influx and membrane depolarization, sensitization of primary nociceptive neurons, and ultimately pain [42,43].

Simultaneously, melittin causes tissue damage and activates pain receptors. Together with MCD, melittin induces mast cells to release mediators, such as histamine, bradykinin, and ATP, which activate G protein-coupled receptors (GPCRs) located on the terminal cell membrane of nociceptive neurons, mediating ligands of one or more signaling pathways (e.g., protein kinase A (PKA), protein kinase C (PKC), phospholipase C (PLC), or extracellular signal-regulated kinase (ERK) [44,45]). Other chemical mediators, including growth factors acting on tyrosine kinase receptors, can activate phospholipase C (PLC), phosphoinositide 3-kinase (PI3K), and mitogen-activated protein kinase (MAPK). Kinases activated by these signaling pathways, in turn, phosphorylate existing protein targets expressed in nociceptive neurons.

Apamin, another critical peptide component of BV, selectively inhibits Ca^2+^-dependent K^+^ channels in the central nervous system. Hence, we speculate that apamin selectively inhibits SK channels (KCNN4, KCNK1, and KCNQ5) on corneal neurons, decreasing delayed hyperpolarization of cells and enhancing sustained neuronal firing, thereby increasing cellular sensitivity to pain. Likewise, it has been reported that MCD blocks Ca^2+^-activated K^+^ channels, thereby increasing neural excitability.

Chen et al. [46] reported that an imbalance between excitatory amino acids (EAAs) and inhibitory amino acids (IAAs) in the spinal cord is associated with the maintenance of persistent pain-related behaviors in inflammatory pain states. PLA2 activates and enhances glutamatergic excitatory synaptic transmission in glial neurons. Spinal substantia gelatinosa neurons not only accept glutamatergic excitatory transmission but also GABAergic and glycinergic inhibitory transmission.

It has been reported [47] that corneal injury can cause corneal cells and locally infiltrated immune cells to release a large number of pro-inflammatory cytokines (such as TNF, IL-6, and IL-1β), resulting in ocular inflammation [48]. In addition, peritoneal injection of BV has been demonstrated to induce neuronal cell death in mice. Furthermore, PLA2 not only directly induces neuronal death in vitro but also indirectly by activating cytokines, such as TNF and IL-1β [49]. Therefore, we speculate that in the CBS model, the injured cornea releases a large number of pro-inflammatory factors, especially TNF, under the synergistic action of melittin and PLA2, causing corneal nerve injury and activating primary sensory neurons, which leads to localized demyelination and axonal degeneration. In the BV group, the expression of injury-related genes *FOS*, *SOCS3*, and *ATF3* was upregulated, indicating that BV injection not only causes a series of immunoinflammatory reactions in the cornea but also injury of primary neurons under the synergism of cytokines.

In the clinic, conventional treatment is generally preferred for CBS injuries, with immediate removal of the venomous stinger and topical steroid medication to control inflammation, with a view to protecting vision; however, some patients experience serious complications and require complex surgery. Here, RNA-seq transcriptomics analysis was performed for the first time on the cornea following the bee sting and Tobradex^®^ application. The results show that the expression levels of pro-inflammatory factors, such as IL-1, IL-6, IL-24, CXCL2, and CXCL3, were significantly downregulated after treatment, indicating that Tobradex^®^ can alleviate a series of immunoinflammatory responses caused by corneal trauma via the cytokine–cytokine receptor interaction by downregulating these pro-inflammatory factors [50].

A PPI network was constructed to identify the hub genes in the corneal healing process following a bee sting, which revealed that *GFAP*, *MAG*, *PLP1*, and *MBP* became hub genes after Tobradex^®^ therapy. *GFAP* is associated with astrocyte differentiation, *MAG* and *MBD* are involved in the formation of myelinated Schwann cells, and *PLP1* is a major component of myelin sheaths. It was discovered that the astrocyte-specific marker *Aldh1l1* and the unmyelinated Schwann cell markers *SCM7A*, *MATN4*, *NCAM1*, and *GFRA3* were considerably upregulated in the BT group. KEGG enrichment analysis of the hub genes also indicates that Tobradex^®^ can protect corneal nerves by repairing axons. It has been reported that treatment of sciatic nerve injury with dexamethasone results in the thickening of nerve fibers and myelin sheathing, thereby accelerating axonal regeneration [51]. Accordingly, we hypothesize that Tobradex^®^ may play a role in corneal nerve repair by “activating” Schwann cells to promote the formation of new myelin sheathing.

In addition, this study had some limitations that could not be ignored. CBS is one of the most common ocular traumas and may lead to blindness because of the complications arising due to the toxic effects of bee venom. However, we investigated only the changes in the anterior segment, especially in the cornea, due to our experimental conditions. Additionally, we did not further verify and explore the results of bioinformatics of high-throughput sequencing.

## 4. Conclusions

In the present study, a stable CBS animal model was successfully established for the first time. Based on this model with or without Tobradex^®^ therapy, we demonstrated that the phenotypic changes and the active components of BV sensitized the nociceptive receptors at the corneal nerve endings and damaged the corneal nerves to some extent. The treatment with Tobradex^®^ exerted a protective effect on the corneal nerves by restoring damaged axons through the upregulated expression of GFAP, MAG, PLP1, and MBP, besides demonstrating a potent anti-inflammatory action.

## 5. Materials and Methods

### 5.1. Animals

C57BL/6 mice (SPF, 20–22 g, male) were purchased from Slack Laboratory Animal Company (Shanghai, China) and housed in the Laboratory Animal Center of Xiamen University (Xiamen, China). Ocular diseases were excluded before experimental treatment. All experiments were carried out based on the Association for Research in Vision and Ophthalmology (ARVO) statement for Use of Animals in Ophthalmology and Vision Research and were approved by the Animal Ethics Committee of Xiamen University.

### 5.2. Preparation of the BV Solution

Western honeybee (*Apis mellifera*)’s BV was purchased as powder from Yimin apiary (China). The powder was dissolved in 1× PBS to prepare a 0.5 mg/mL BV solution, which is the minimum concentration that can cause a severe corneal bee sting.

### 5.3. Establishment of the Corneal Bee Sting Model

A total of 45 mice were randomly assigned to 5 groups with 9 in each: the normal group (Norm), vehicle group (VE), vehicle with therapy group (VT), BV injection group (BV), and BV injection with therapy group (BT). Mice were anesthetized by intraperitoneal injection of 1% pentobarbital sodium solution (5 μL/g body weight), following which 0.5% proparacaine hydrochloride eye drops were applied to numb the cornea, and an eye speculum was used to expose the ocular surface. As shown in Figure 8, under the stereomicroscope, a 32G needle was used to make a tunnel through the central cornea of the right eye into the corneal stroma. In the BV group, 2 μL 0.5 mg/mL BV solution was injected into the corneal stroma with a 34G needle, and the conjunctival sac was washed three times with 1× PBS immediately afterwards. In the BT group, Tobramycin Dexamethasone eye ointment (Tobradex^®^, Alcon Laboratories Inc., Fort Worth, TX, USA) was applied based on the same process as the BV group. The VE group was injected with 2 μL 1× PBS solution, and dexamethasone eye ointment was applied to the VT group based on the same process as the VE group. The eyelids were closed to protect the ocular surface. Finally, the anesthetized mice were placed on a thermostatic electric blanket to recover from the anesthesia.

### 5.4. Slit-Lamp Microscopic Examination

At 6 h, 12 h, 24 h, and 72 h after the injection, the ocular changes (including corneal opacity, iris pigment changes, and pupil size) and sodium fluorescein staining (corneal epithelial defects) were observed using a slit-lamp microscope (BQ-900; Haag-Streit AG, Koeniz, Switzerland). The fluorescent area in the image was analyzed using Fiji [52] to assess epithelial injury.

### 5.5. Optical Coherence Tomography

Optical coherence tomography (OCT) was applied to the biological measurement of the cornea, anterior chamber angle, lens, and other anterior segment structures (RT-100; Optovue, Fremont, CA, USA) at 6 h, 12 h, 24 h, and 72 h after injection. OCT images of the cornea and anterior chamber were analyzed using Fiji to assess the changes.

### 5.6. Preparation of Corneal Whole-Mounts

Mice were sacrificed by cervical dislocation at 24 h and 72 h after the establishment of the model. The eyeballs were quickly fixed in 4% paraformaldehyde solution at 4 °C for 1 h. Subsequently, the whole cornea was cut into four petals with a scalpel under a surgical microscope. The cornea was placed in an EP tube with 1× PBS solution and rinsed three times in a shaker (at slow speed) at room temperature for 10 min each. The cornea was fixed in cold acetone for 2 min, rinsed three times with 1% TD-Buffer for 10 min each, and blocked in 2% BSA solution for 1 h.

### 5.7. F-Actin Staining of Corneal Whole-Mounts

Alexa Fluor™ 555 Phalloidin (A34055, ThermoFisher Scientific, Waltham, MA, USA) was used to observe the overall shape and structure of the cells in the corneal whole-mount according to the method provided by ThermoFisher Scientific. Nuclei were labeled with Hoechst33342 (62249, ThermoFisher Scientific, USA). Whole-mounts were observed using a ZEISS LSM 880 confocal microscope. The images were used to create a 3D reconstruction of the cornea with IMARIS 9.2.0.

### 5.8. TUNEL Assay of Corneal Whole-Mounts

A colorimetric TUNEL apoptosis assay kit (C1088, Beyotime, Shanghai, China) was used to observe the level of apoptosis in corneal whole-mounts according to the method provided by Beyotime. Nuclei were labeled with Hoechst33342. Whole-mounts were observed using a ZEISS LSM 880 confocal microscope.

### 5.9. Whole-Genome mRNA Sequencing and Bioinformatics Analysis

Total RNA was isolated as described previously [53]. RNA was segmented by interrupting buffer, the random N6 primers were reverse-transcribed, and cDNA was synthesized. The synthetic double-stranded DNA was flattened and phosphorylated at the 5′ end to form a sticky end protruding an “A” at the 3′ end, and then connected to a bulbous connector with a protruding “T” at the 3′ end. The ligand products were amplified by PCR using specific primers. PCR products were thermally denatured into single strands, following which the single-stranded DNA was cycled with a bridge primer to obtain a single-stranded circular DNA library. The cDNA library of all specimens in this experiment was collected and sequenced with an Illumina Hi Seq X-Ten (Huada Gene, Shenzhen, China). Hub genes were screened out comprehensively by the Cyto-scape software 3.7.2 in combination with the cyto-Hubba plug-in through 12 topological analysis methods.

### 5.10. Statistical Analysis

All the experiments were conducted using the randomized double-blind control method, and each experiment was repeated at least three times. Experimental data are represented as the mean ± standard deviation. GraphPad Prism 9 (San Diego, CA, USA) was used for statistical analysis and data expression. PCA was performed to detect the degree of aggregation in the samples. HCA was performed to cluster genes with similar expression patterns. Statistical analysis and visualization were performed in R version 3.6.3 and ggploT2 packages (for visualization).

## Figures and Tables

**Figure 1 toxins-14-00468-f001:**
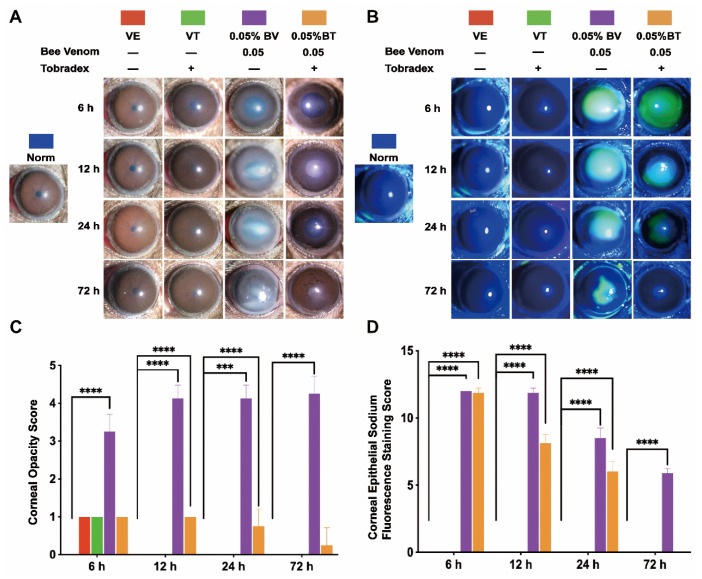
Slit-lamp microscopic validation and observation of the corneal bee sting (CBS) model. (**A**,**C**) Representative images and statistical analysis of the corneal opacity changes in different groups (Norm = 0 ± 0, VE = 1 ± 0, VT = 1 ± 0, BV = 3.25 ± 0.46, BT = 1 ± 0 at 6 h; Norm = 0 ± 0, VE = 0 ± 0, VT = 0 ± 0, BV = 4.13 ± 0.35, BT = 1 ± 0 at 12 h; Norm = 0 ± 0, VE = 0 ± 0, VT = 0 ± 0, BV = 4.13 ± 0.35, BT = 0.75 ± 0.46 at 24 h; Norm = 0 ± 0, VE = 0 ± 0, VT = 0 ± 0, BV = 4.25 ± 0.46, BT = 0.25 ± 0.46 at 72 h). (**B**,**D**) Representative images and statistical analysis of the corneal epithelial defects observed by sodium fluorescein staining (Norm = 0 ± 0, VE = 0 ± 0, VT = 0 ± 0, BV = 12 ± 0, BT = 11.88 ± 0.35 at 6 h; Norm = 0 ± 0, VE = 0 ± 0, VT = 0 ± 0, BV = 11.88 ± 0.35, BT = 8.13 ± 0.64 at 12 h; Norm = 0 ± 0, VE = 0 ± 0, VT = 0 ± 0, BV = 8.50 ± 0.76, BT = 6.00 ± 0.76 at 24 h; Norm = 0 ± 0, VE = 0 ± 0, VT = 0 ± 0, BV = 5.88 ± 0.35, BT = 0 ± 0 at 72 h). Results are shown as means ± SD of eight different experiments, *** *p* < 0.001, **** *p* < 0.0001.

**Figure 2 toxins-14-00468-f002:**
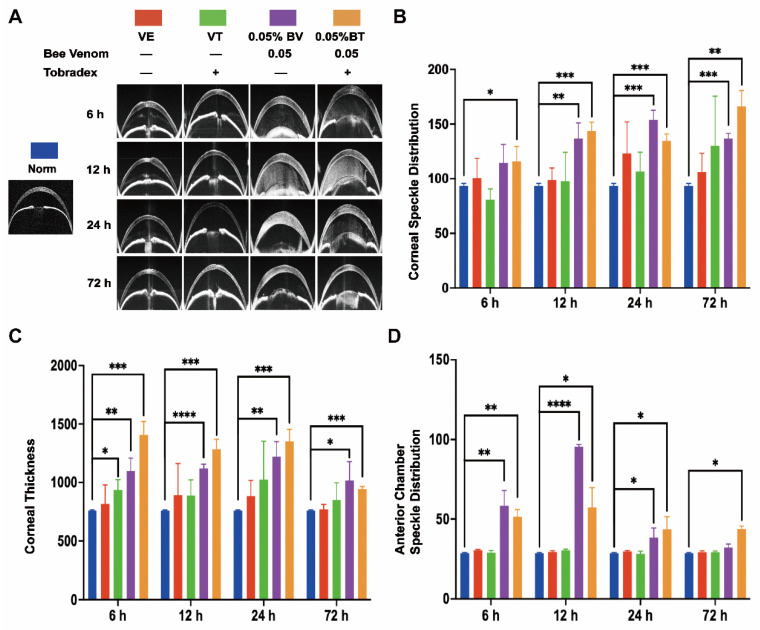
Anterior segment optical coherence tomography (AS-OCT) observation of the corneal bee sting (CBS) model. (**A**) Representative images of the AS-OCT observation in different groups. (**B**) Corneal edema evaluated by corneal speckle distribution (Norm = 93.64 ± 2.04, VE = 100.42 ± 18.14, VT = 80.63 ± 10.06, BV = 114.32 ± 17.04, BT = 115.93 ± 13.46 at 6 h; Norm = 93.64 ± 2.04, VE = 98.75 ± 11.01, VT = 97.84 ± 26.37, BV = 136.61 ± 14.36, BT = 143.54 ± 8.12 at 12 h; Norm = 93.64 ± 2.04, VE = 123.18 ± 28.77, VT = 106.46 ± 17.81, BV = 153.96 ± 8.63, BT = 134.72 ± 6.11 at 24 h; Norm = 93.64 ± 2.04, VE = 106.08 ± 17.25, VT = 129.97 ± 45.68, BV = 136.61 ± 4.84, BT = 166.11 ± 14.57 at 72 h). (**C**) Corneal cross-sectional area (Norm = 760.33 ± 5.03, VE = 818.33 ± 161.23, VT = 937.00 ± 88.50, BV = 1099.33 ± 108.30, BT = 1405.67 ± 117.56 at 6 h; Norm = 760.33 ± 5.03, VE = 892.67 ± 271.32, VT = 889.33 ± 134.66, BV = 1120.00 ± 39.04, BT = 1284.33 ± 87.32 at 12 h; Norm = 760.33 ± 5.03, VE = 885.00 ± 113.74, VT = 1026.00 ± 325.88, BV = 1222.67 ± 125.53, BT = 1350.00 ± 104.86 at 24 h; Norm = 760.33 ± 5.03, VE = 770.00 ± 43.86, VT = 849.33 ± 147.45, BV = 1018.67 ± 160.28, BT = 945.00 ± 21.70 at 72 h). (**D**) Anterior chamber exudates evaluated by anterior chamber speckle distribution (Norm = 28.60 ± 0.44, VE = 30.62 ± 0.59, VT = 28.94 ± 1.65, BV = 58.28 ± 9.74, BT = 51.38 ± 4.65 at 6 h; Norm = 28.60 ± 0.44, VE = 29.46 ± 0.86, VT = 30.59 ± 0.82, BV = 95.38 ± 1.35, BT = 57.24 ± 12.58 at 12 h; Norm = 28.60 ± 0.44, VE = 29.82 ± 0.69, VT = 28.22 ± 1.81, BV = 38.53 ± 5.85, BT = 43.49 ± 7.89 at 24 h; Norm = 28.60 ± 0.44, VE = 29.25 ± 0.94, VT = 29.25 ± 0.90, BV = 32.34 ± 1.95, BT = 43.67 ± 2.04 at 72 h). Results are shown as means ± SD of three different experiments, * *p* < 0.05, ** *p* < 0.01, *** *p* < 0.001, **** *p* < 0.0001.

**Figure 3 toxins-14-00468-f003:**
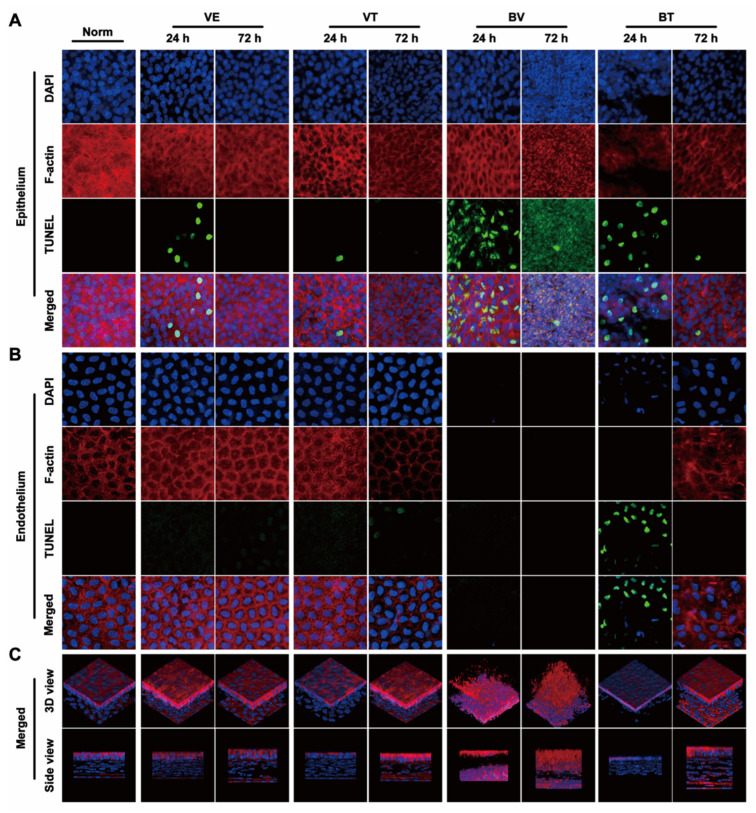
Fluorescence laser confocal microscopic observation of whole-mount cornea. (**A**) Representative images of changes in corneal epithelial nuclei (Blue: DAPI) and cytoskeleton (Red: F-actin) as well as apoptosis (Green: TUNEL) in different groups. (**B**) Representative images of changes in the corneal endothelial nuclei (Blue: DAPI) and cytoskeleton (Red: F-actin) as well as apoptosis (Green: TUNEL) in different groups. (**C**) Representative images of 3D reconstruction (Blue: DAPI, Red: F-actin) of the corneas in different groups.

**Figure 4 toxins-14-00468-f004:**
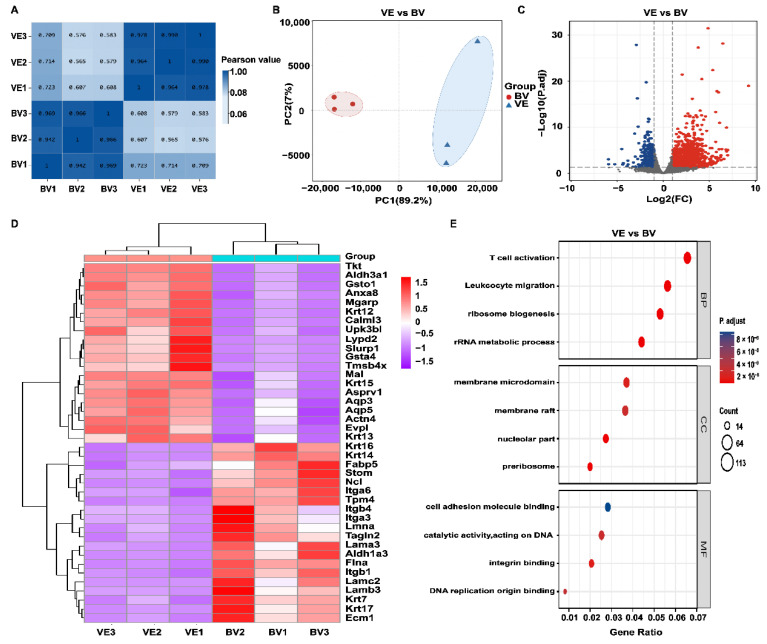
Identification of differentially expressed genes (DEGs) between the VE and BV groups. (**A**) Correlation analysis of VE vs BV. (**B**) Principal component analysis (PCA) of VE vs BV. (**C**) Volcano plot of VE vs BV. Blue dots represent downregulated genes; gray dots represent genes with no significant difference; red dots represent upregulated genes. (**D**) Hierarchical cluster analysis (HCA) of metabolic genes. Data are shown for fragments transcribed per kilobase per million mapped reads (FPKM). Red: upregulated expression; Purple: downregulated expression. (**E**) Functional enrichment analysis of upregulated genes in the VE and BV groups. Top: Gene Ontology (GO) of biological processes (BP), cellular components (CC), and molecular functions (MF) are shown (Log2 fold change > 1, FDR < 0.01), respectively.

**Figure 5 toxins-14-00468-f005:**
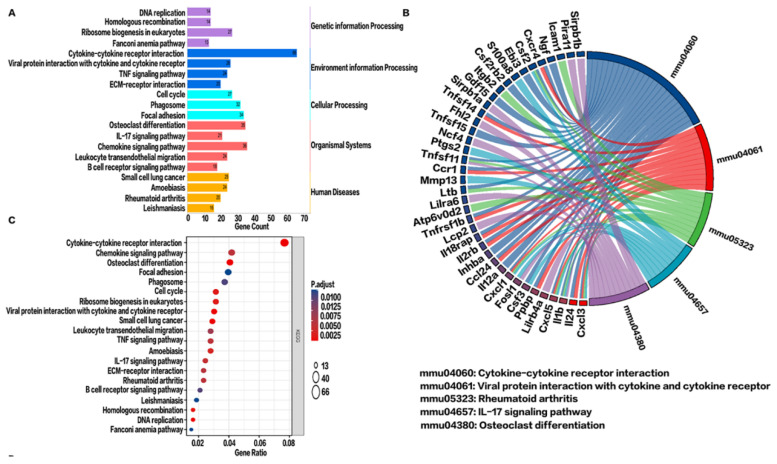
Signaling pathway analysis of upregulated genes in the VE and BV groups. (**A**) Classification of upregulated differential gene signaling pathways. (**B**) Kyoto Encyclopedia of Genes and Genomes (KEGG) string diagram. (**C**) Enrichment analysis of the top 20 pathways of upregulated differential gene KEGG pathways. Here, *p* values less than 0.01 after adjustment confer the most significance.

**Figure 6 toxins-14-00468-f006:**
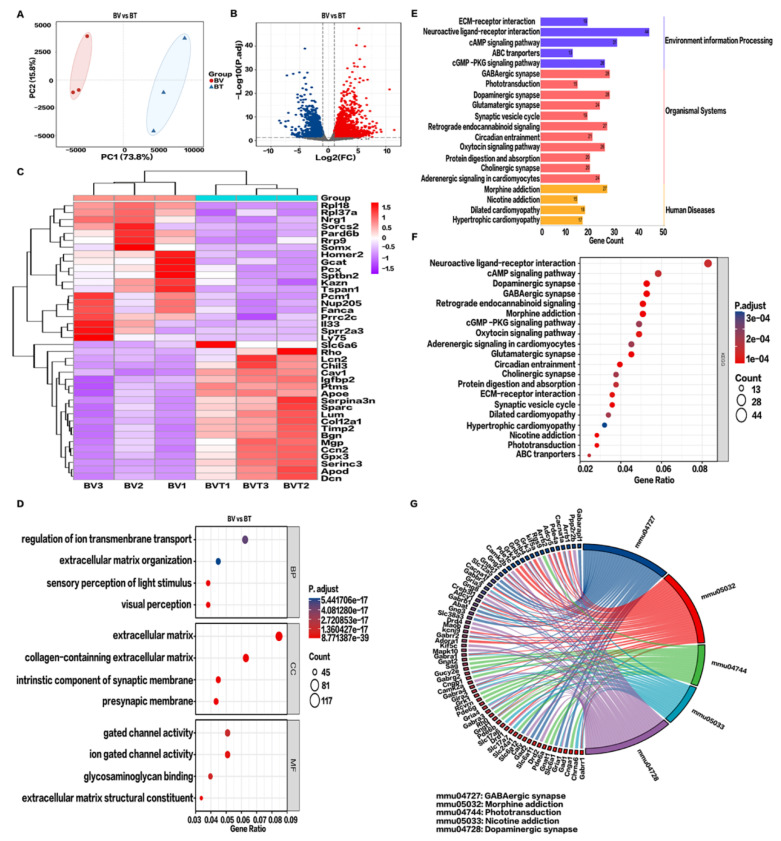
Transcriptomics analysis after Tobradex^®^ treatment. (**A**) Principal component analysis (PCA) of BV vs BT. (**B**) Volcano plot of BV vs BT. Blue dots represent downregulated genes; gray dots represent genes with no significant difference; red dots represent upregulated genes. (**C**) Hierarchical cluster analysis (HCA) of metabolic genes. Data show fragments per kilobase of transcript per million mapped reads (FPKM). Red: upregulated expression, Purple: downregulated expression. (**D**) Functional enrichment analysis of BV vs BT. Biological processes (BP), cellular components (CC), and molecular functions (MF), Log2 FC > 1, FDR < 0.01. (**E**) Kyoto Encyclopedia of Genes and Genomes (KEGG) signaling pathway categories. (**F**) Enrichment analysis of the top 20 pathways of upregulated differential gene KEGG pathways in BV vs BT. *p* values less than 0.01 after adjustment confer the most significance. (**G**) KEGG string diagram of the top five enriched signaling pathways. Colored lines show genes linked to the indicated pathways. Sorting is based on observed log2 FC values. Adjacent red boxes show decreasing intensity.

**Figure 7 toxins-14-00468-f007:**
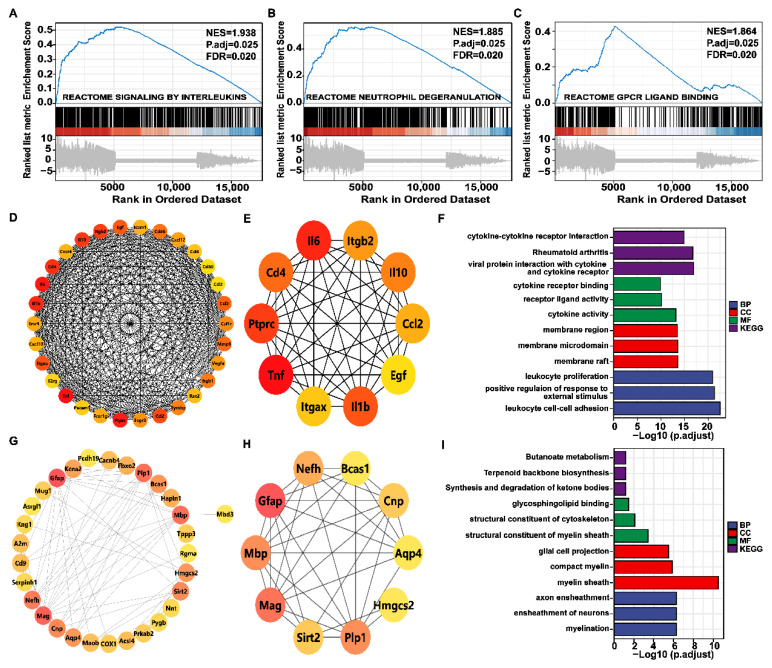
Gene set enrichment analysis (GSEA) and protein–protein interaction (PPI) network. (**A**–**C**) Top 3 GSEA of gene sets for the corneal bee sting. Positive and negative normalized enrichment scores (NESs) indicate higher and lower expression, respectively. False discovery rate (FDR) (q value) < 0.25 and p.adjust < 0.05 confers statistical significance. (**D**) PPI network of the top 30 hub genes in the BV group. The redder the color, the higher the degree. (**E**) PPI network of the top 10 hub genes in the BV group. (**F**) Gene Ontology (GO) and Kyoto Encyclopedia of Genes and Genomes (KEGG) analysis of the hub gene in the BV group. (**G**) The PPI network of the top 30 hub genes in the BT group. (**H**) The PPI network of the top 10 hub genes in the BT group. (**I**) GO and KEGG analysis of the hub gene in the BT group.

**Figure 8 toxins-14-00468-f008:**
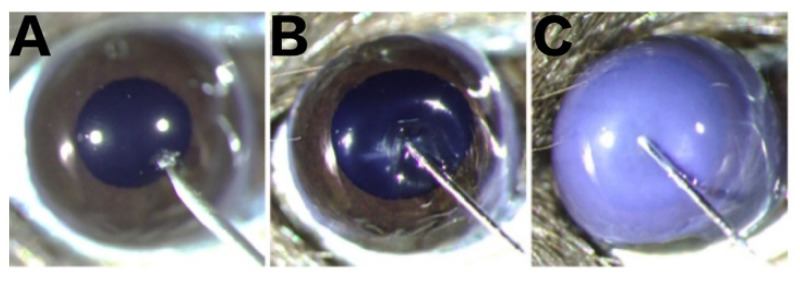
Intrastromal injection. (**A**) Tunnel penetration into the corneal stroma using a sterile disposable syringe fitted with a 32G needle. (**B**) Successful tunnel establishment before intra-stromal injection using a microinjector fitted with a 34G needle. (**C**) Successful intra-stromal injection.

## Data Availability

The datasets generated and analyzed in the current study are available from the corresponding author on reasonable request.

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
