# Peer review of "Phenotypic and Transcriptomics Analyses Reveal Underlying Mechanisms in a Mouse Model of Corneal Bee Sting"

_toxins, 2022, doi:10.3390/toxins14070468_

Round 1
Reviewer 1 Report
This study investigated 65 the pathological characteristics and pathogenesis of CBS of animal model and provided new therapeutic strategies for the clinical treatment.
I think that this paper was well-written and worth to be reported.
I have some minor comments:
1. In methods part, there were five groups; though there were only four groups in some figures.
2. Please state the limitation of this study.
3. It seems to be hidden, though, please state the IRB approval if it was not written in this manuscript.
Author Response
Thank you for your kind comments and suggestions, we have responded to your valuable comments as follows.
1. In methods part, there were five groups; though there were only four groups in some figures.
Response: In Figures 1-3, the graph of the normal group (Norm) was easily overlooked. Also, in Figures 4-7, we did not analyze all the groups together but only compared some of them.
2. Please state the limitation of this study.
Response. Thanks. It has been stated in the last paragraph of the Discussion.
4. It seems to be hidden, though, please state the IRB approval if it was not written in this manuscript.
Response: Thanks. It has been stated in detail in the manuscript.
Reviewer 2 Report
Abstract
L 18. “ ...but also reduced corneal pain...”
How have the authors studied the reduction in pain in the present investigation?
Tobradex is a trading name, I suggest using a trademark symbol when the above is used in the manuscript.
L31 ... and even death”.
Including the following reference may be appropriate.
Feás, X.; Vidal, C.; Remesar, S. What We Know about Sting-Related Deaths? Human Fatalities Caused by Hornet, Wasp and Bee Stings in Europe (1994–2016). Biology 2022, 11, 282. https://doi.org/10.3390/biology11020282
FIg. 1 & 2. Check a typo error: “Bee Vomen”.
The figures should be fully understood without the need to refer to the main text. In this sense, I suggest that all acronyms used in the figures be specified in the figure legend itself.
Figure 4a has an excessively small font size. It is necessary to increase the font size to allow its reading. The same comment for Figure 5d.
L277-279.
It would be desirable to provide some references regarding these clinical cases.
It is necessary to correct the style of the manuscript. Different fonts are used. References must also be consistent with the style of the journal.
Author Response
Thank you for your kind comments and suggestions. We apologize for our negligence. Also, we have responded to your valuable comments as follows.
Abstract
L 18. “ ...but also reduced corneal pain...”
How have the authors studied the reduction in pain in the present investigation?
Response: Thanks. We studied the reduction in pain by noting the changes in pain-related genes and signaling pathways.
Tobradex is a trading name, I suggest using a trademark symbol when the above is used in the manuscript.
Response: Thanks. They have been corrected.
L31 ... and even death”.
Including the following reference may be appropriate.
Feás, X.; Vidal, C.; Remesar, S. What We Know about Sting-Related Deaths? Human Fatalities Caused by Hornet, Wasp and Bee Stings in Europe (1994–2016). Biology 2022, 11, 282. https://doi.org/10.3390/biology11020282
Response: Thanks. The reference has been added.
FIg. 1 & 2. Check a typo error: “Bee Vomen”.
The figures should be fully understood without the need to refer to the main text. In this sense, I suggest that all acronyms used in the figures be specified in the figure legend itself.
Figure 4a has an excessively small font size. It is necessary to increase the font size to allow its reading. The same comment for Figure 5d.
Response: Thanks. They have been corrected.
L277-279.
It would be desirable to provide some references regarding these clinical cases.
Response: Thanks. The references have been added.
It is necessary to correct the style of the manuscript. Different fonts are used. References must also be consistent with the style of the journal.
Response: Thanks. They have been corrected.
Reviewer 3 Report
The article is interesting and valuable. Bee venom (BV) is usually associated with pain since, when humans are stung by bees, local inflammation and even an allergic reaction. Especially, dangerous seems corneal bee sting. As Authors mentioned, CBS may lead to blindness. Therefore, the mechanisms of bee venom toxic effects should be explained.
I would like to underline, that the Authors paid attention to this aspect, but they should explain these mechanisms clearly. They should pay attention especially on cytokines action during ocular inflammation. They described it on page 11, but they should complete these informations and present them on figure.
We know, that bee venom consists of a mixture of substances, principally of proteins
and peptides, including enzymes as well as other types of molecules in a very low concentration, but it is very important to indicate which components are responsible for the side effects and showing the synergism action of them. The Authors mentioned about components in introduction, but they missed references. In my opinion, they should discuss the contribution of the individual components of bee venom to the side effects of a bee sting on a cornea.
I am under impression of this article. The conducted study on animal model was very well planned, but should be better described.
The reviewer suggests major revisions. The list of suggestions and remarks are listed below:
Point 1: In the Abstract, TUNEL assay should be explain. Moreover, in line 16 Authors indicate TNF. In line 330 they mentioned TNF-α. It should be clarified in all text. In line 17, Authors should better describe composition of Tobradex ( tobramycin together with dexamethasone) and indicate their concentrations in this ointment.
In line 13, Authors should clarify stromal thickening…. what?
Point 2: In the Introduction section, there is no information of references, lines 36, 45
Moreover, in line 41 k+ channels should be written as K+ channels.
In addition in this section, in lines 50-51 Authors should explain the variations in the composition of bee venom.
Point 3: In the Results section, figures should be corrected as below:
In Figure 1 Authors should correct bee venom.
In Figure 2 Authors should explain meaning of stars and show the values of ±SD.
In Figure 3 Authors should improve the images. They should have better resolution.
Figure 4 should have better resolution especially A.
Figure 5 D is completely unreadable. It should be corrected.
Figure 6 should be also corrected.
In line 253 should be TNF-α or TNF, please explain
Point 4: In Discussion section in line 306, Authors should explain the abbreviation of PLA2-LOX.
In line 307, they should insert references.
In line 348, Authors should describe these mechanisms.
Additionally, Authors should explain the basis on which their hypothesis, mentality was created.
Point 5: The font should be standardized throughout the text.
Point 6: In statistical analysis, Authors should mention about HCA and PCA analysis.
Point 7: The Conclusions section summarize the results. Chapter Conclusions may be modified. The conclusions should underline the authors' hypothesis.
Author Response
Thank you for your kind comments and suggestions. We apologize for our negligence. Also, we have responded to your valuable comments as follows.
Point 1: In the Abstract, TUNEL assay should be explain. Moreover, in line 16 Authors indicate TNF. In line 330 they mentioned TNF-α. It should be clarified in all text. In line 17, Authors should better describe composition of Tobradex (tobramycin together with dexamethasone) and indicate their concentrations in this ointment.
In line 13, Authors should clarify stromal thickening…. what?
Response: Thanks. They have been corrected.
Point 2: In the Introduction section, there is no information of references, lines 36, 45
Moreover, in line 41 k+ channels should be written as K+ channels.
In addition to in this section, in lines 50-51 Authors should explain the variations in the composition of bee venom.
Response: Thanks. They have been corrected.
Point 3: In the Results section, figures should be corrected as below:
In Figure 1 Authors should correct bee venom.
In Figure 2 Authors should explain meaning of stars and show the values of ±SD.
In Figure 3 Authors should improve the images. They should have better resolution.
Figure 4 should have better resolution especially A.
Figure 5 D is completely unreadable. It should be corrected.
Figure 6 should be also corrected.
Response: Thanks. The figure problems have been corrected. If Figure 5D is still difficult to read, please confirm if it needs to be placed in the supplementary materials. The Hi-Res Figure 5D is at the end of this file.
In line 253 should be TNF-α or TNF, please explain
Response: TNF-α has been mentioned in previous studies. Our analysis focused only on TNF but could not judge its subtype. We have changed TNF-α to TNF to improve the logic of the manuscript.
Point 4: In Discussion section in line 306, Authors should explain the abbreviation of PLA2-LOX.
In line 307, they should insert references.
Response: Thanks. Necessary corrections have been made.
In line 348, Authors should describe these mechanisms.
Response: Thanks. The mechanisms have been described.
Additionally, Authors should explain the basis on which their hypothesis, mentality was created.
Response: Thanks. The basis has been explained.
Point 5: The font should be standardized throughout the text.
Response: Thanks. It has been corrected.
Point 6: In statistical analysis, Authors should mention about HCA and PCA analysis.
Response: Thanks. HCA and PCA analyses have been mentioned.
Point 7: The Conclusions section summarize the results. Chapter Conclusions may be modified. The conclusions should underline the authors' hypothesis.
Response: Thanks. This section has been modified.
Round 2
Reviewer 3 Report
Respected Authors
Thank you for your corrected version. In my opinion, it is necessary to improve article again.
In line 73 and in line 88 before reference Authors should insert space
In figures 1 and 2 Authors have written still bee vemon against bee venom
Figure 5 is not enough clear, Authors should add it as supplement
In lines 694-700 Authors should add PCA and HCA analyses
In introduction, Authors should explain the variety of bee venom better. The composition depends on the species, bee age, geopgraphical localization etc. Authors can find more information in publication : Bee venom in wound healing ; https://doi.org/10.3390/molecules26010148
Author Response
Respected Reviewer,
Thank you for your kind comments and suggestions, we have revised and explained your valuable comments and questions as follows:
In line 73 and in line 88 before reference Authors should insert space
Response: Thanks. They have been corrected.
In figures 1 and 2 Authors have written still bee vemon against bee venom
Response: Much to our regret, they have been corrected.
Figure 5 is not enough clear, Authors should add it as supplement
Response: Thanks. It has been added as Figure S1 in the Supplemental Materials section.
In lines 694-700 Authors should add PCA and HCA analyses
Response: Thanks. They have been added.
In introduction, Authors should explain the variety of bee venom better. The composition depends on the species, bee age, geographical localization etc. Authors can find more information in publication: Bee venom in wound healing; https://doi.org/10.3390/molecules26010148
Response: Thanks. They have been better explained.